# Rewards Simplified: Reducing Risk in RL for Cyber Defence

## Abstract

Recent years have seen an explosion of interest in autonomous cyber defence agents trained to defend computer networks using deep reinforcement learning. These agents are typically trained in cyber gym environments using dense, highly engineered reward functions which combine many penalties and incentives for a range of (un)desirable states and costly actions. Dense rewards help alleviate the challenge of exploring complex environments but risk biasing agents towards suboptimal and potentially riskier solutions, a critical issue in complex cyber environments. We thoroughly evaluate the impact of reward function structure on learning and policy behavioural characteristics using a variety of sparse and dense reward functions, two well-established cyber gyms, a range of network sizes, and both policy gradient and value-based RL algorithms. Our evaluation is enabled by a novel ground truth evaluation approach which allows directly comparing between different reward functions, illuminating the nuanced inter-relationships between rewards, action space and the risks of suboptimal policies in cyber environments. Our results show that sparse rewards, provided they are goal aligned and can be encountered frequently, uniquely offer both enhanced training reliability and more effective cyber defence agents with lower-risk policies. Surprisingly, sparse rewards can also yield policies that are better aligned with cyber defender goals and make sparing use of costly defensive actions without explicit reward-based numerical penalties.

## 1 Introduction

Cyber attacks are increasingly frequent and sophisticated, straining limited cyber defence resources and threatening critical digital systems that people depend upon worldwide. There has been a rising level of interest in using machine learning (ML) methods to improve cyber security; in particular deep reinforcement learning (DRL) which has the ability to learn complex policies from interaction alone, enabling the discovery of strategies unconstrained by flawed system or security models. DRL based autonomous cyber defence (ACD) agents, which have gathered much attention in the literature, could discover novel techniques and provide automation for tasks that currently occupy human analysts.

Cyber gyms provide efficient and controlled environments for ACD agents. This is particularly important for network security tasks, enabling the large number of interactions required for training without risking production networks or systems. Accordingly, many cyber gyms have been created to enable training agents that defend networked systems (Vyas et al., 2023). Cyber gyms define one or more Markov Decision Processes (MDPs) in terms of a state space comprising network and host information, an action space of defensive activities, and a reward function aligned to defensive objectives. ACD reward functions are typically highly engineered based on human judgment, combining multiple penalties and incentives determined for a variety of defensive actions and network states (Andrew et al., 2022; Standen et al., 2021). Dense rewards may be preferable because of expedited learning, providing apparently effective solutions using fewer environment steps during training, but they also risk constraining agents to sub-optimal solutions (Riedmiller et al., 2018). This is especially concerning for ACD agents which might then contain avoidable weaknesses that are difficult to identify in advance of an attack. Furthermore, dense rewards draw potentially arbitrary numerical equivalences between network states and actions. As the scale and complexity of cyber tasks grow this becomes increasingly challenging to manage and the risks of undesirable agent behaviour are exacerbated.

At the expense of generally requiring more training iterations, sparse rewards place fewer constraints on the solution space and could enable preferable or more effective policies to be discovered. Existing work has not investigated the possibility that dense rewards might limit the performance of ACD agents trained using DRL. To investigate this possibility, and summarising the main contributions of this work, we: (1) propose a ground truth scoring mechanism for network security cyber gyms which allows a direct comparison between agents trained using different reward functions, (2) evaluate a comprehensive range of sparse and dense reward functions using two popular cyber gyms which are adapted to illustrate our ground truth mechanism, and (3) show that sparse reward functions can enhance the effectiveness, reliability and risk-profiles of ACD agents across a variety of network sizes and topologies, action spaces, MDP models and DRL algorithms.

## 2 BACKGROUND

Here we provide an introduction to ACD, motivate evaluating ACD agents more accurately, and define the key metrics we later build upon to fully evaluate the impact of reward functions in ACD.

### 2.1 AUTONOMOUS CYBER DEFENCE

ACD agents aim to actively mitigate attacks on computer networks using ML techniques rather than traditional rule-based approaches. By alleviating the bottlenecks of human response speed and information processing, ACD agents could provide a much needed counterbalance to the ever-increasing scale and sophistication of cyber threats. Reinforcement learning (RL), and particularly DRL given the enormity of data generated by computer networks, is particularly promising as it allows learning defensive strategies from interaction alone without the need for explicit models of how networks, systems, and attackers behave. Such models must continually be updated as attackers evolve, frequently undermining the tools and techniques that derive security proofs or assurances from their correctness. By observing the network state and choosing defensive actions, DRL agents can learn novel and adaptive strategies for defending computer networks that do not depend on potentially incorrect or outdated assumptions.

Since their learning is guided by maximising long-term rewards, ACD agents critically depend on the rewards provided throughout training. Furthermore, the exploration required for learning from trial-and-error demands a cyber gym allowing extensive experimentation (i.e., risk-taking) without jeopardising valuable production systems. Many cyber gyms have been created (Vyas et al., 2023), provided publicly (Microsoft, 2021; Oesch et al., 2024; Andrew et al., 2022), and even used for competitions seeking the best performing agents (Standen et al., 2021; Hicks et al., 2023; Foley et al., 2022). Despite these promising developments, previous work on ACD is limited to evaluating performance using only mean episodic rewards, and variance of the same, over a number of fixed-policy rollouts. Unlike games (e.g., chess) which correspond relatively naturally to the MDP framework, defending a network of computer hosts does not. Real-world attackers are not confined to turn-based interactions, partial observability affects many aspects of the network, and there is never a state where the defender can be definitively crowned the winner.

Most cyber gyms, and prominent ACD competitions, have hand-crafted dense reward functions that are used to train and evaluate agents. Such rewards may misrepresent the true performance of agents and it is impractical for them to accurately represent human knowledge (Hu et al., 2020), biasing models towards possibly lower-performance and higher-risk strategies. There is a need, which we illustrate and address for the first time to the best of our knowledge, for evaluation methods that accurately represent the ground truth of complex cyber environments. Our ground truth scoring mechanism permits a direct and reproducible comparison between different reward strategies, enabling experiments that empirically quantify the performance and risk characteristics of reward functions in ACD environments.

### 2.2 RELIABILITY AND RISK IN RL

The reliability and risks of RL agents is a critical issue, especially for cyber defence applications where inconsistent performance can be costly or dangerous. Training reliability metrics measure how consistently an RL algorithm performs across multiple training runs, and risk metrics quantify expectations of worst-case performance.

TRAINING RELIABILITY

To evaluate the impact of reward function on training reliability in ACD agents, we build upon the quantitative RL training reliability metrics proposed by Chan et al. (2020) based on dispersion variability i.e., the width of the mean episodic rewards distribution.

**Dispersion variability across time (DT)**  measures the stability of RL training across time. Smooth monotonic policy improvements offer the lowest $DT$ scores, indicating high reliability during training and lowered computational costs. $DT$ is measured by averaging the inter-quartile range (IQR) within a sliding window along each detrended training curve. Detrending ensures positive trends in policy improvement do not influence the metric and is calculated using differencing (i.e., $y'_t = y_t - y_{t-1}$). Where $I$ denotes the total number of runs, the average DT across multiple runs is calculated:

$$\bar{\text{DT}} = \frac{\sum_{i=0}^{I} \text{DT}_i}{I}$$

**Dispersion variability across runs (DR)**  measures the reproducibility of RL training across multiple runs. Low DR indicates high consistency between training runs, meaning fewer total training runs are required to discover the best performing agents. DR is measured by averaging IQR across multiple training runs at each evaluation step, ensuring the metric captures differences resulting from random initialisation or environment stochasticity. Let $\bar{R}_i$ denote the mean episodic reward over some window of training run $i$, and $\{\bar{R}_1, \bar{R}_2, \ldots, \bar{R}_I\}$ the set of all such $\bar{R}_i$ across $I$ total runs, then:

$$\text{DR} = \text{IQR}\big(\{\bar{R}_i\}_{i=1}^{I}\big)$$

RISK AFTER TRAINING

In ACD we are particularly concerned about the worst-case scenarios for a given agent. We calculate this by considering the worst-case expected loss across multiple rollouts of each trained policy.

**Conditional Value at Risk (CVaR)**  quantifies the risk associated with worst-case scenarios, defined by some quantile $\alpha$, i.e., expected performance in the worst $\alpha$ fraction of cases (Acerbi & Tasche, 2002). By focussing on extreme values in the tails of the distribution, CVaR complements IQR methods in which they are cut off to focus on dispersion between central quartiles.

**Risk across Fixed-Policy Rollouts (RF)**  is calculated by applying CVaR to the distribution of multiple fixed-policy evaluation rollouts. Where $X = \{\bar{R}_1, \bar{R}_2, \ldots, \bar{R}_I\}$ denotes the set of mean episodic returns from the trained policy, and $\text{VaR}_\alpha$ the $\alpha$ quantile of $X$, then:

$$\text{RF}_\alpha(X) = \text{CVaR}_\alpha(X) = \mathbb{E}[X \mid X \leq \text{VaR}_\alpha(X)]$$

## 3 METHODOLOGY

Here we outline the methodology and experimental setup used to evaluate how different reward functions impact agent performance and training reliability in ACD.

### 3.1 YAWNING TITAN CYBER GYM

Yawning Titan (YT) (Andrew et al., 2022) is a well-established cyber gym providing an abstract, graph based network simulation environment for training defensive (blue) agents to defend a network by minimising the number of compromised nodes. To establish foundational insights, and to minimise variance and implementation errors in the first instance, we configured YT to simulate a linear network structure with a fixed entry node for the attacking (red) agent which follows a fixed lateral-movement strategy aiming to compromise as many nodes as possible.

The YT observation space comprises a vector embedding the network adjacency matrix and both the vulnerability and compromise status of each node. We set the vulnerability of each node to 1, conservatively modelling the most powerful red agent whose attacks never fail. We create two action spaces: (1) basic – with two actions: "scan network" and "restore node", and (2) extended – which also adds "place decoy". The place decoy action is a proactive defence replicating the use of a

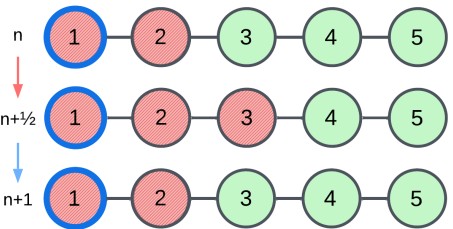

Figure 1: One step of the YT network environment illustrating an intra-step node compromise that is concealed by standard cyber gym evaluations.

deceptive "canary", a technique sometimes used to detect and delay attackers in real world networks. The red action space has two actions: "do nothing" and "basic attack", where the fixed red policy is to at random perform a basic attack 90% of the time and do nothing otherwise (10%).

### 3.2 CYBER AUTONOMY GYM FOR EXPERIMENTATION (CAGE)

Cyber Autonomy Gym for Experimentation (CAGE) 2 (Kiely et al., 2023) is one of the most popular single-agent ACD environments designed to enable training defensive RL agents in simulated network attack scenarios (Standen et al., 2022; Vyas et al., 2023). Adding considerable complexity in contrast to YT, CAGE 2 defines an enterprise network with 3 subnets and 13 hosts in total: the user subnet with 5 hosts, the enterprise subnet with 3 hosts and an isolated defender host, and the operational subnet with 4 hosts. The network is separated by firewalls such that red agents must compromise multiple hosts to move from user subnet hosts, via the enterprise subnet, to the operational target. The observation space is a vector of 52 bits, comprising 4 bits detailing state and adversary information for each host. The action space includes 6 high-level actions (sleep, monitor, analyse, remove, restore, decoy) which are expanded to detail type and target for a total of 145 different actions.

In our experiments we use the refined CAGE 2 implementation, miniCAGE (Emerson et al., 2024), which eliminates bugs and increases training speeds but otherwise has exactly the same environment dynamics, red agent behaviour, observation and action spaces, and network topology. Of the two red agents included in CAGE 2, we use the "b-line" attacker, which uses partial prior knowledge of the network to exploit the shortest path from entry node to impacting the operational target.

### 3.3 GROUND TRUTH

To the best of our knowledge, previous work on ACD is limited to evaluating performance using the mean episodic reward and its variance over a large number of rollouts. This assumes the MDP model captures the "ground truth", and that the episodic reward is aligned with preferred ACD goals. However, cyber gyms are highly complicated environments which simulate both red and blue agent actions. According to the MDP framework actions are taken during discrete time steps, requiring a determined order in which red and blue actions occur. Current cyber gyms overlook this crucial detail and choose either a fixed order or prioritise actions according to some arbitrary function.

Illustrated in Figure 1, one issue with the MDP framework's requirement for discrete time steps is that the observation provided at the end of each step can omit critical network events occurring intra-step which are resolved before the reward is determined. Concretely–red agents may compromise nodes during the step, just before the blue agent removes the compromise, and this will not be reflected in the reward or observation returned to the agent. This makes it impossible for agents to reliably distinguish between states in which nodes have been compromised and those in which no compromise occurred. Consequently, prior ACD evaluation metrics fail to distinguish between agents with potentially very different ground truth behaviour.

**Ground Truth Score** ($\mathrm{Score_{GT}}$)  To overcome the limitations of discrete step-wise evaluation in cyber gyms we introduce the ground truth score, $\mathrm{Score_{GT}}$, calculated as the maximum ($\max$) number of compromised nodes over both the intra- and end-step. In general, where $m_t^{(\mathrm{intra})}$ and $m_t^{(\mathrm{end})}$ are the intra- and end-step number of compromised nodes, respectively:

$$\mathrm{Score_{GT}}(t) = \max\left(m_t^{\mathrm{intra}}, m_t^{\mathrm{end}}\right) \tag{1}$$

For Figure 1, $\text{Score}_{\text{GT}}(\cdot) = \max(3, 2) = 3$ i.e., capturing the ground truth that 3 nodes were compromised during the time step. The ground truth score provides a more accurate measure of agent performance that is independent of agent order and does not depend on the reward function used during training—enabling the impact of reward on agent performance to be evaluated robustly.

### 3.4 Evaluating reliability across different rewards

To evaluate the impact of reward function on training reliability using a single risk metric, we introduce a normalised version of Chan et al. (2020)'s DR measure (defined in Section 2.2). To capture variability in converged performance rather than early fluctuations we restrict our application to the final 20% of steps. For each training run $i$ we calculate the mean episodic reward $R_i$ across the final 20% of training steps. Then, we apply mean normalisation to each run's mean episodic reward:

$$R_i' = (R_i - \mu)/\sigma$$

Across $I$ total training runs, our normalised DR metric is calculated as the IQR over the mean normalised mean episodic rewards:

$$\text{DR}' = \text{IQR}(R_i' \mid \forall i \in I)$$

### 3.5 Experiments

Our experiments evaluate the performance, risk, and reliability of the different reward functions defined in Table 1. These reward structures are representative of both the complex, dense reward functions currently used by most cyber gyms including YT and CAGE, and an encompassing range of sparse rewards aligned with the goal of defending the network by minimising the number of compromised nodes. The sparse reward functions place fewer constraints on the optimisation objective, e.g., by avoiding numerical comparison between nodes and defensive actions, thus might enable agents to learn more effective policies. Note that we use the terms "positive" and "negative" principally to refer to the goal of mitigating adversarial node compromise in the network. The most positive outcome is that the network is at full-health and zero nodes are compromised. Similarly, the most negative scenario entails the complete compromise of all network nodes. See also Appendix K.

In both YT and CAGE, our evaluation applies the ground truth score and reliability metrics defined in Section 3.3 and 3.4. Furthermore, upper and lower RF refer to the bounds of the average per-step ground truth score at risk across rollouts (see Section 2.2) determined by the $\alpha = 0.05$ quartile. All experiments are trained for 25 independent runs and the final policies are evaluated for 1000 episodes, resulting in a $\text{Score}_{\text{GT}}$, upper and lower RF, DT, DR$'$ and 95% confidence intervals (CI) for each network size, reward function, and agent order. Agent order is fixed for each corresponding training run and evaluation. We did not search for optimal hyperparameters in this work as the Stable-Baselines3 defaults (see Appendix B) proved sufficient in both PPO and DQN, however tuned hyperparameters may further enhance learning in any given experiment. Experiments were run using Intel i9 and Apple M1 and M3 Pro CPUs, alongside NVidia RTX 4090 GPUs, requiring 720 processor days in total for the results in this paper. Including additional preliminary experiments and experimental re-runs the total rises to 1100 processor days.

Table 1: The sparse and dense reward functions evaluated.

| Reward Type | YT Reward per time step | CAGE Reward per time step |
|---|---|---|
| Sparse Positive (SP) | +1 if no nodes compromised only. | +1 if no nodes compromised and red agent is in user subnet. |
| Sparse Negative (SN) | -1 if all nodes compromised only. | -1 if operational server is impacted. |
| Sparse Positive-Negative (SPN) | +1 if none and -1 if all nodes compromised, respectively. | +1 if no nodes compromised and red agent is in user subnet, and -1 if operational server is impacted. |
| Dense Negative (DN) | -1 per compromised node. | N/A. |
| Complex Dense Negative (CDN) | Action penalties and -1 per compromised node, see Appendix A. | Standard CAGE 2 reward, see Appendix A. |

YAWNING TITAN EXPERIMENTS

Informed by insights provided by our ground truth mechanism, we trained agents using three different orderings of red and blue actions: red then blue (standard in YT and CAGE), blue then red, and random. The random order performs an alternating sequence of red then blue, and blue then red, with the initial order randomised in each episode. The random order includes the worst-case for the defender where the red agent acts twice consecutively before blue can act. These experiments evaluate the relationship between reward structure and robustness to inter-step agent order. Prior work utilising CAGE has shown that environment complexity and the inability to interpret agent behaviour scales rapidly as network size grows (Foley et al., 2022). Complex network simulations obfuscate the relationship between reward function and final policy outcomes. Thus, we begin in YT with the least complex sub-problem: 2 nodes and 2 actions (basic) and iteratively scale the network size up to 50 nodes before then including the proactive decoy action (extended). These experiments evaluate the impact of reward structure as both the network size and action space are scaled up.

In all experiments the episode length is fixed at 100 steps and each agent is trained using PPO, one of the most widely used algorithms for training ACD agents (Vyas et al., 2023). To demonstrate that our findings are not specific to PPO we also perform additional experiments using DQN (Mnih et al., 2015) (see Appendix D). To ensure convergence during training, we scale the number of training steps so that for network sizes of 2, 5, 10, 20, and 50 nodes, agents are trained for 0.5, 1, 1.5, 2, and 2.5 million steps, respectively.

CAGE EXPERIMENTS

To explore the generalisability of our findings to non-linear network structures and expanded state-action spaces we also trained agents in the MiniCAGE environment using the set of rewards detailed in Table 1. The episode length was fixed at 100 steps and we trained agents using both PPO and DQN for 2.5 million timesteps (see Appendix J for DQN results).

## 4 RESULTS

Here we present key results showing how reward structure impacts performance, risk and reliability.

SP AND SPN REWARDS PERFORM BEST ON AVERAGE

Providing an overarching view of the results in YT, shown in Table 2, we consolidate the ground truth scores, risk, and training reliability of each reward function averaged across all network sizes and agent orders. The SPN reward function achieves the best scores: fewer nodes are compromised

Table 2: PPO results in YT, for the extended action space, averaged across all network sizes and agent orders for sparse positive (SP), sparse negative (SN), sparse positive negative (SPN), dense negative (DN) and complex dense negative (CDN) reward functions.

| Reward Function | Score$_{GT}$ | Average Evaluation Reliability | | | | 95% CI | |
|---|---|---|---|---|---|---|---|
| | | Lower RF | Upper RF | $\bar{DT}$ (e-3) | DR$'$ | LL | UL |
| SP | 2.69 | 2.46 | 2.87 | 0.11 | 0.12 | 2.02 | 3.36 |
| SN | 10.29 | 9.00 | 10.90 | 0.09 | 0.17 | 9.10 | 11.47 |
| SPN | **2.00** | 1.82 | 2.16 | 0.08 | 0.19 | 1.38 | 2.63 |
| DN | 6.29 | 5.84 | 6.60 | 2.33 | 0.12 | 5.14 | 7.44 |
| CDN | 6.21 | 5.71 | 6.52 | 2.45 | 0.31 | 5.10 | 7.32 |

Table 3: Results for PPO agents trained in MiniCAGE using 4 reward functions: sparse positive (SP), sparse negative (SN), sparse positive negative (SPN) and the default CAGE reward function (CDN).

| Reward Function | Score$_{GT}$ | Average Evaluation Reliability | | | | 95% CI | |
|---|---|---|---|---|---|---|---|
| | | Lower RF | Upper RF | $\bar{DT}$ (e-3) | DR$'$ | LL | UL |
| SP | **1.29** | 0.97 | 3.11 | 0.34 | 0.46 | 1.24 | 1.34 |
| SN | 2.77 | 1.85 | 3.64 | 0.05 | 0.19 | 2.66 | 2.87 |
| SPN | 1.35 | 0.97 | 2.93 | 0.36 | 0.47 | 1.23 | 1.48 |
| CDN (default CAGE rewards) | 1.41 | 1.06 | 2.02 | 0.55 | 0.31 | 1.31 | 1.51 |

on average than agents trained using any other reward function. SP rewards provide the next-best performing agents, followed by DN, CDN and finally SN rewards. All of the sparse reward functions, including SN, show significantly lower DT than the dense rewards, confirming greater training reliability across time (albeit to a low average performance for SN). Reliability across runs is two orders of magnitude higher for dense rewards in both action spaces indicating greatly reduced reproducibility. Across every YT configuration, as shown in Tables 4, 5 and 6, the best performing PPO agents result from either either SP or SPN reward functions. This is also true for DQN agents as shown in Appendix D. Similarly in the CAGE environment, see Table 3 and Appendices J), the SP reward function achieves the best $\text{Score}_{GT}$. Both SP and SPN rewards outperform the standard CAGE reward function in terms of $\text{Score}_{GT}$, and the upper 95% confidence interval for SP is lower than the average $\text{Score}_{GT}$ of the standard CAGE reward function.

PERFORMANCE AND RISK SCALING WITH NETWORK SIZE

Evaluating performance as the network size increases shows how each reward function scales to larger, and therefore more realistic, state-action spaces. In YT we evaluate trained agents in networks with 2, 5, 10, 20 and 50 nodes, averaging scores over all runs and agent orders for each network size. Table 5 shows the average ground truth performance, and worst 5% percent of performances i.e., risk, for all agent orders. As network size increases, the performance and risk differences between reward functions widens. In the smallest 2 and 5 node networks, both SPN and SP reward functions yield the best agents with closely matched average performance and worst-case risks–especially in the basic action space (see Appendix C). In the largest two network sizes the advantages of SPN rewards are magnified, providing significantly better policies with correspondingly reduced risks. For 10 node networks there is an exception to the overall trend where SP rewards outperform SPN in the extended action space. As discussed further in Section 5, a closer analysis of the data reveals this is likely because, in the extended action space, both SP and SPN rewards enable learning optimal strategies for defending networks when the agent order is blue then red. The results show that SP and SPN rewards not only perform best overall but also scale favourably as state-action spaces increase.

Table 4: YT PPO agent performance and risk evaluation scores across all network sizes for the extended action space. Results are averaged over all agent orders for each reward function.

| | Evaluation across network sizes | | | | | | | | | |
| | 2 | | 5 | | 10 | | 20 | | 50 | |
| Reward Function | Score GT | Upper RF | Score GT | Upper RF | Score GT | Upper RF | Score GT | Upper RF | Score GT | Upper RF |
|---|---|---|---|---|---|---|---|---|---|---|
| SP | **0.60** | 0.64 | **0.62** | 0.66 | **0.63** | 0.67 | 1.87 | 1.96 | 9.75 | 10.44 |
| SN | 1.19 | 1.26 | 3.59 | 3.82 | 7.47 | 7.92 | 12.43 | 13.28 | 26.76 | 28.20 |
| SPN | 0.92 | 0.98 | 0.97 | 1.01 | 0.85 | 0.89 | **0.69** | 0.73 | **6.58** | 7.17 |
| DN | 0.98 | 1.03 | 1.28 | 1.41 | 3.21 | 3.42 | 8.19 | 8.45 | 17.78 | 18.70 |
| CDN | 0.85 | 0.90 | 8.73 | 9.23 | 4.03 | 4.18 | 8.46 | 8.73 | 16.06 | 17.02 |

Table 5: YT PPO agent results for each agent action order combination in the extended action space, averaged over all network sizes, for each reward function.

| | Red then Blue | | | Blue then Red | | | Random | | |
| Reward Function | $\text{Score}_{GT}$ | Upper RF | CI UL | $\text{Score}_{GT}$ | Upper RF | CI UL | $\text{Score}_{GT}$ | Upper RF | CI UL |
|---|---|---|---|---|---|---|---|---|---|
| SP | **0.90** | 0.96 | 0.90 | **0.27** | 0.28 | 0.72 | 6.91 | 7.38 | 8.47 |
| SN | 9.31 | 9.95 | 10.92 | 9.01 | 9.51 | 10.72 | 12.54 | 13.23 | 12.77 |
| SPN | **0.90** | 0.96 | 0.90 | 0.61 | 0.63 | 1.23 | **4.50** | 4.89 | 5.75 |
| DN | 4.13 | 4.31 | 5.74 | 2.98 | 3.08 | 4.17 | 11.75 | 12.42 | 12.40 |
| CDN | 5.65 | 5.80 | 5.29 | 3.99 | 4.11 | 4.14 | 13.24 | 14.13 | 12.52 |

THE IMPACT OF AGENT ORDER

Evaluating the impact of different agent orders on reward function performance reveals how the real-world constraints of uncertain attacker timing and dynamics could impact the performance and worst-case risks of ACD agents. Table 4 shows the average agent performance across all runs and network sizes in each of the three agent orders: red then blue, blue then red, and random. Continuing the trend, SP and SPN have considerably higher performance scores and lower risks than the dense

rewards. When the agent order is randomised, the scores for all reward functions are greatly reduced, highlighting the sensitivity of DRL-based ACD agents to adversary timing. Notably, SPN significantly outperforms the other reward functions when the agent order is random–the most challenging and realistic setting in which the agent order cannot be assumed before an episode begins. Furthermore, when the agent order is blue then red and agents use the extended action space (i.e., blue can place decoys and moves before red), the average performance for SP agents reaches 0 meaning an ideally secure network with no compromised nodes during any episode. Collectively these results showcase the strong inter-relationships between reward function, action space, and performance risks when agent timing cannot be anticipated. See Appendix E for the average agent $Score_{GT}$ alongside mean episodic rewards for each reward function in YT.

Table 6: Agent order results for YT agents trained in the 50 node network, extended action space.

| Reward Function | Red then Blue | | | Blue then Red | | | Random | | |
|---|---|---|---|---|---|---|---|---|---|
| | $Score_{GT}$ | Best $Score_{GT}$ | No. of Optimal Runs (/25) | $Score_{GT}$ | Best $Score_{GT}$ | No. of Optimal Runs (/25) | $Score_{GT}$ | Best $Score_{GT}$ | No. of Optimal Runs (/25) |
| SP | **0.90** | 0.90 | 25 | 1.36 | 0.00 | 23 | 26.98 | 0.94 | * |
| SN | 22.84 | 3.93 | 0 | 24.33 | 1.99 | 0 | 33.10 | 27.00 | * |
| SPN | **0.90** | 0.90 | 25 | **0.81** | 0.00 | 24 | **18.04** | 0.94 | * |
| DN | 12.53 | 1.89 | 0 | 8.42 | 0.90 | 0 | 32.39 | 27.67 | * |
| CDN | 10.05 | 1.89 | 0 | 6.71 | 1.89 | 0 | 31.44 | 27.34 | * |

\* The optimal policy score is non-trivial so we do not count the number of optimal runs

## 5 DISCUSSION

Empirical results for network sizes ranging from 2 to 50 nodes, irrespective of attacker timing or whether proactive actions are available, confirm that SP and SPN rewards provide the best performing blue agents with minimised worst-case risks. Where the optimal scores (i.e., 0.9 for red then blue and 0 for blue then red agent orderings) were computed analytically, a fine-grained evaluation of these metrics for the largest network we evaluate (shown in Table 6) further corroborates this result. Additional training curves for the 50 node network can be seen in Appendix I. Here, both SPN and SN reward functions uniquely enable agents to learn optimal strategies which limit the attacker to very few, and even 0 in favourable conditions, compromised nodes.

In CAGE, SP and SPN agents also performed best, achieving the lowest $Scores_{GT}$. This shows our results generalise to more representative networks with multiple subnets and complex non-linear behaviours including hosts with different vulnerabilities. To understand why sparsely rewarded policies perform better we analyse the behaviour in Appendix G. While SP agents result in slightly elevated operational server impacts in comparison to the standard CAGE rewards, 0.24 vs. 0.02 average per episode; there are significantly fewer successful operational and enterprise privilege escalations, 0.25 and 1.29 vs. 7.89 and 22.22 average per episode, respectively. In addition to much lower overall privileged host access counts (22.75 vs. 34.63 average per episode), SP agents confine 21 of these (92.31%) to user subnet hosts. This policy is much better aligned to network security objectives as user hosts have the fewest network privileges, and the least overall impact on operations.

Also in Appendix G, we confirm that sparsely rewarded agents use the costly restore action more sparingly, and with greater focus on the user hosts, than agents trained using the standard CAGE rewards. Given the lack of numerical penalty for these actions in SP and SPN, this result highlights that sparse rewards can avoid riskier, less aligned policies which may otherwise result from incorrectly translating human domain insights into numerical rewards. Our results demonstrate that dense reward functions can suboptimally constrain the performance of ACD agents, introducing avoidable risks and reducing training reliability across runs. Furthermore, our ground truth scoring mechanism and its application in this work illustrates the importance of more considerate evaluation in current cyber gym environments. Many cyber gyms fail to capture important inter-step agent behaviours (e.g., including compromised nodes), obscuring crucial performance and risk differences between policies.

Dense reward functions, which are standard practice in cyber gyms, risk artificially constraining the performance of ACD agents and weakening the resulting security of networks they defend. More broadly, our results show that ACD agents require a reward function to provide sufficient reward signal (i.e., "can be encountered frequently during training") and goal-alignment. Since dense rewards introduce bias, sparse rewards are indicated for goal-alignment. However, sparse rewards can also

present exploration problems as their frequency during training is highly task dependent. The sparse positive rewards utilised here are both sparse in action-state space, and can be encountered frequently provided some uncompromised network state(s) can be identified. It remains for future work to understand the scaling limitations of this approach in real world networks, but the SN reward illustrates an additional challenge faced by sparse ACD reward structures. Specifically, as the defensive policy improves throughout training, the network becomes less frequently (entirely) compromised and correspondingly provides less reward signal from which to learn further improvements. These findings likely have applications in many other cyber defence tasks beyond network defence, for example in web application vulnerability discovery (Lee et al., 2022; Al Wahaibi et al., 2023). Whilst our work is intended to advance the cyber defence domain, a potential negative impact is the dual use from adversaries who could seek to use it for malicious purposes. As cyber gyms increase in realism, moving ACD agents closer to operational environments, it is essential to establish and empirically validate the design of effective, efficient and risk-reducing reward functions.

## 6 RELATED WORK

Reward functions, and how best to formulate them, has been widely discussed in relation to the emergence of intelligent behaviour within the RL framework (Silver et al., 2021; Vamplew et al., 2022). Many real-world RL applications including robotics (Dorigo & Colombetti, 1994) and video games (OpenAI et al., 2019; Song et al., 2019) utilise reward shaping to address sample inefficiency, aiming to guide learning towards useful policies by incorporating domain knowledge to reduce the learning problem difficulty. Reward shaping also arises when gradient-based methods are used to augment extrinsic rewards, such as the adaptive utilisation of a reward shaping function (Hu et al., 2020), or to provide "intrinsic motivation" towards uncertainty-reducing actions (Pathak et al., 2017). Nevertheless, a requirement for policy invariance is that reward shaping functions must apply the difference of an arbitrary potential function between successive states (Ng et al., 1999). Any other reward transformation may bias resulting trained policies away from the optimal solution (Riedmiller et al., 2018). This work establishes for the first time, with implications for widely-used cyber gyms, the performance, risk and training efficiency implications of reward function design in ACD.

Prior work has sought to benchmark RL algorithm performance (Duan et al., 2016) assess algorithm reliability (Henderson et al., 2018), and measure policy reliability during and after training (Chan et al., 2020). Whilst these methods help to evaluate an agent trained using a specific reward function, comparing multiple reward functions remains challenging, and often task-specific, because episodic-reward-based evaluation crucially lacks an external frame of reference. DRL has been used for a variety of real-world cyber security tasks including alert prioritisation (Tong et al., 2020), language model "jailbreak" prompt optimisation (Chen et al., 2024), fuzzing compilers (Li et al., 2022), finding web application vulnerabilities (Lee et al., 2022; Al Wahaibi et al., 2023), finding cache timing attacks (Luo et al., 2023), and overcoming hardware trojan detection methods (Gohil et al., 2022). For a broader survey on RL-based ACD we refer readers to Vyas et al. (2023). The closest previous work (Bates et al., 2023) investigates 4 different reward shaping approaches (normalised, linearly scaled, non-linear scaling and curiosity-based exploration (Pathak et al., 2017)) in the standard CAGE environment. In contrast to this work, their results are inconclusive, policies are evaluated using only episodic rewards, and no consideration is given to DRL algorithm, agent order, policy risks, training reliability, or the effects of scaling network size or action spaces.

## 7 CONCLUSION

In this work we introduce a novel ground truth scoring method and addresses a key shortcoming of cyber gyms: neglecting intra-step node compromises when evaluating agent performance. This work enables a more accurate, risk-aware, and comprehensive evaluation of ACD policies, independent of the training reward structure or agent-timings. Through extensive experiments in YT and CAGE, two well-established cyber gyms, we show that agents trained with simpler, sparse reward functions outperform those trained on conventional dense rewards and maintain higher reliability across increasing network sizes. Notably, our SPN reward function yields policies with significantly fewer compromised nodes in worst-case scenarios, especially when attacker timing is randomised (i.e., the most realistic setting). Our findings underscore the great importance of reward functions and their relationship to risk and goal alignment in cyber environments. Lastly, we have highlighted the complex inter-relationships between reward functions, action spaces, network size, and attacker timings, relating them to the ground truth performance of ACD agents.

## 8 ETHICS STATEMENT

Our work fully adheres to the guidelines articulated in the ICLR Code of Ethics. In the introduction, we motivate the work by discussing the need for effective autonomous cyber defence considering society's dependence on cyber systems and the growing complexity of attacks, highlighting the corresponding societal benefits. The main focus of this work is constructing more effective autonomous cyber defence agents. Furthermore, the environments we adapt for our experimental method are abstract representations and, even should they be adapted for offensive purposes, will not yield agents capable of attacking real-world networks.

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

## A  COMPLEX DENSE NEGATIVE REWARD FUNCTION

The complex dense negative (CDN) reward function is charitably (i.e., we favourably interpret the spirit of these rewards rather than focusing on specific flaws) derived from the heavily shaped reward functions used by several of the most popular cyber gyms including CAGE's CybORG, PrimATE and Yawning Titan (YT) (Standen et al., 2022; Andrew et al., 2022). A typical "real world" CDN-type reward function, combining both negative penalties and positive rewards for various blue agent actions and environment states, is taken from the YT GitHub repository and partially described below in Table 7 below. The full YT reward function includes additional rewards with detailed nuances and caveats not shown here for clarity of presentation. Although somewhat intuitive, the specific YT reward values are arbitrary and no clear justification is provided for the magnitudes and inevitable numerical equivalences assigned.

We constructed the CDN and CD reward functions to charitably represent, with decreasing complexity and shaping, the reward functions found in leading cyber gyms. Some rewards were considerably simplified, for example the penalties for compromised node states, and others were omitted entirely because our experimental designs exclude the actions altogether. Our newly introduced decoy action was assigned an arbitrary penalty based on the insight that restoring a node entirely would clearly be more disruptive than temporarily disturbing one node service.

Table 7: Action- and state-level shaping terms for the *YT*, *CDN* and *DN* reward functions.

| | Reward Function | | |
|---|---|---|---|
| | **YT** | **CDN** | **DN** |
| **Actions** | | | |
| Reduce Vulnerability | $-0.5$ | — | — |
| Restore Node | $-1$ | $-0.5$ | $0$ |
| Make Node Safe | $-0.5$ | — | — |
| Scan Network | $0$ | $0$ | $0$ |
| Isolate Node | $-10$ | — | — |
| Connect Node | $0$ | — | — |
| Add Deceptive Node | $-8$ | — | — |
| Place Decoy | — | $-0.25$ | $0$ |
| Do Nothing | $+0.5$ | $-0.1$ | $0$ |
| **States** | | | |
| Network compromise / vulnerability | *>30% nodes compromised:* $-1$ per compromised node *Vulnerability reduced:* $+4\times$ reduction | $-1$ per compromised node | $-1$ per compromised node |

## B  HYPERPARAMETERS FOR TRAINING

Here we present the hyperparameters used for training in both the YT and MiniCAGE environments.

Table 8: Hyperparameters for PPO models

| Hyperparameter | Value |
|---|---|
| Learning Rate | $3 \times 10^{-4}$ |
| Number of Hidden Layers | 2 |
| Hidden Layer Size | 64 |
| GAE Lambda | 0.95 |
| Clip Range | 0.2 |
| Gamma | 0.99 |
| Value Function Coefficient | 0.5 |
| Number of Epochs | 10 |
| Batch Size | 64 |

Table 9: Hyperparameters for DQN models, using the default Stable-Baselines3 hyperparameters with the exception of the buffer size (changed from $1e^6$ to $200,000$) and final epsilon (changed from $0.05$ to $0.005$)

| Hyperparameter | Value |
|---|---|
| Learning Rate | $1 \times 10^{-4}$ |
| Batch Size | 32 |
| Gamma | 0.99 |
| Train/Update Frequency | 4 |
| Buffer Size | 200,000 |
| Exploration Initial Epsilon | 1 |
| Exploration Final Epsilon | 0.005 |

## C  BASIC ACTION SPACE RESULTS IN YAWNING TITAN

Here we detail our YT results using the basic action space comprising (1) scan network and (2) restore node.

Table 10: PPO results (Basic Action Space) averaged across all network sizes and agent orders for sparse positive (SP), sparse negative (SN), sparse positive negative (SPN), dense negative (DN) and complex dense negative (CDN) reward functions.

| Reward Function | $Score_{GT}$ | Average Evaluation Reliability | | | | 95% CI | |
|---|---|---|---|---|---|---|---|
| | | Lower RF | Upper RF | DT (e-3) | DR′ | LL | UL |
| SP | 4.58 | 4.16 | 4.88 | 0.07 | 0.19 | 3.83 | 5.32 |
| SN | 9.92 | 8.90 | 10.69 | 0.05 | 0.21 | 8.82 | 11.03 |
| SPN | 1.97 | 1.75 | 2.42 | 0.09 | 0.26 | 1.47 | 2.46 |
| DN | 5.84 | 5.42 | 6.16 | 2.98 | 0.29 | 4.73 | 6.04 |
| CDN | 6.03 | 5.61 | 6.37 | 2.90 | 0.39 | 5.21 | 6.86 |

Table 11: PPO agent performance and risk evaluation scores across network sizes (Basic Action Space). Results are averaged over all agent orders for each of the 5 reward functions.

| | Evaluation across network sizes | | | | | | | | | |
|---|---|---|---|---|---|---|---|---|---|---|
| | 2 | | 5 | | 10 | | 20 | | 50 | |
| Reward Function | Score GT | Upper RF | Score GT | Upper RF | Score GT | Upper RF | Score GT | Upper RF | Score GT | Upper RF |
| SP | 1.05 | 1.11 | 1.10 | 1.17 | 2.05 | 2.22 | 3.88 | 4.01 | 14.81 | 15.32 |
| SN | 1.23 | 1.29 | 3.72 | 3.90 | 6.87 | 7.22 | 12.08 | 13.27 | 25.72 | 26.16 |
| SPN | 1.05 | 1.11 | 1.23 | 1.97 | 1.05 | 1.11 | 1.23 | 1.31 | 5.30 | 5.88 |
| DN | 1.16 | 1.22 | 1.26 | 1.32 | 3.39 | 3.52 | 8.26 | 8.51 | 15.13 | 16.10 |
| CDN | 1.31 | 1.36 | 8.03 | 8.28 | 4.23 | 4.35 | 7.58 | 7.83 | 14.70 | 15.84 |

Table 12: PPO agent results for each agent action order combination (Basic Action Space), averaged over all network sizes, for each of the 5 reward functions. CI UL is the upper limit of the 95% confidence interval.

| | Red then Blue | | | Blue then Red | | | Random | | |
|---|---|---|---|---|---|---|---|---|---|
| Reward Function | $Score_{GT}$ | Upper RF | CI UL | $Score_{GT}$ | Upper RF | CI UL | $Score_{GT}$ | Upper RF | CI UL |
| SP | 0.90 | 0.96 | 0.90 | 5.11 | 4.99 | 6.64 | 7.73 | 8.35 | 8.43 |
| SN | 8.02 | 8.71 | 9.69 | 9.55 | 9.43 | 10.93 | 12.20 | 12.96 | 12.46 |
| SPN | 0.90 | 0.96 | 0.90 | 1.11 | 1.54 | 1.28 | 3.90 | 4.34 | 5.21 |
| DN | 3.22 | 3.30 | 3.93 | 3.10 | 3.10 | 4.09 | 11.19 | 12.00 | 11.76 |
| CDN | 3.80 | 3.87 | 4.24 | 4.73 | 4.86 | 4.36 | 12.97 | 13.87 | 11.98 |

Table 13: Detailed PPO results for agents trained in the 50 node network (Basic Action Space).

| | 50 Node Network Evaluation | | | | | | | | |
| | Red then Blue | | | Blue then Red | | | Random | | |
| Reward Function | $Score_{GT}$ | Best $Score_{GT}$ | No. of Optimal Runs (/25) | $Score_{GT}$ | Best $Score_{GT}$ | No. of Optimal Runs (/25) | $Score_{GT}$ | Best $Score_{GT}$ | No. of Optimal Runs (/25) |
|---|---|---|---|---|---|---|---|---|---|
| SP | 0.90 | 0.90 | 25 | 12.68 | 0.90 | 2 | 30.86 | 25.02 | 0 |
| SN | 19.72 | 1.89 | 0 | 26.48 | 1.89 | 0 | 30.97 | 27.06 | 0 |
| SPN | 0.90 | 0.90 | 25 | 14.31 | 0.90 | 3 | 13.57 | 1.34 | 0 |
| DN | 7.68 | 0.90 | 2 | 8.30 | 1.89 | 0 | 29.40 | 24.41 | 0 |
| CDN | 6.36 | 0.90 | 4 | 8.77 | 1.89 | 0 | 28.96 | 27.07 | 0 |

# D  DQN RESULTS IN YAWNING TITAN

Here are results for the extended action space trained using the DQN algorithm.

Table 14: DQN results averaged over network sizes 2 to 50, and all agent orders, for all 5 reward functions: sparse positive (SP), sparse negative (SN), sparse positive negative (SPN), dense negative (DN) and complex dense negative (CDN).

| Reward Function | $Score_{GT}$ | Average Evaluation Reliability | | | | 95% CI | |
| | | Lower RF | Upper RF | $\bar{DT}$ (e-3) | $DR'$ | LL | UL |
|---|---|---|---|---|---|---|---|
| **Extended Action Space** | | | | | | | |
| SP | 1.48 | 0.88 | 2.38 | 0.70 | 0.44 | 1.03 | 1.93 |
| SN | 5.98 | 4.34 | 7.16 | 0.17 | 0.38 | 3.11 | 8.86 |
| SPN | 1.12 | 0.78 | 1.50 | 0.75 | 0.44 | 0.56 | 1.69 |
| DN | 3.80 | 3.16 | 4.22 | 5.83 | 0.46 | 2.81 | 4.78 |
| CDN | 5.44 | 4.83 | 5.91 | 4.17 | 0.43 | 4.27 | 6.62 |

Table 15: DQN performance and risk scores as network size increases. Results are averaged over all agent orders for all 5 reward functions: sparse positive (SP), sparse negative (SN), sparse positive negative (SPN), dense negative (DN) and complex dense negative (CDN).

| | Evaluation across network sizes | | | | | | | | | |
| | 2 | | 5 | | 10 | | 20 | | 50 | |
| Reward Function | Score GT | Upper RF | Score GT | Upper RF | Score GT | Upper RF | Score GT | Upper RF | Score GT | Upper RF |
|---|---|---|---|---|---|---|---|---|---|---|
| **Extended Action Space** | | | | | | | | | | |
| SP | 0.71 | 0.78 | 0.71 | 0.78 | 0.95 | 1.13 | 2.28 | 3.72 | 2.76 | 5.51 |
| SN | 0.98 | 1.06 | 1.99 | 2.35 | 2.77 | 3.36 | 8.81 | 10.65 | 15.37 | 18.39 |
| SPN | 0.72 | 0.78 | 0.71 | 0.78 | 0.78 | 0.89 | 0.73 | 0.96 | 2.68 | 4.07 |
| DN | 0.69 | 0.76 | 1.01 | 1.11 | 1.25 | 1.42 | 4.88 | 5.43 | 11.14 | 12.40 |
| CDN | 0.69 | 0.76 | 0.92 | 0.99 | 1.11 | 1.29 | 3.86 | 4.28 | 20.63 | 22.23 |

Table 16: DQN results for all agent order combinations, averaged over all network sizes, for all 5 reward functions: sparse positive (SP), sparse negative (SN), sparse positive negative (SPN), dense negative (DN) and complex dense negative (CDN). CI UL is the upper limit of the 95% confidence interval.

| Reward Function | Red then Blue | | | Blue then Red | | | Random | | |
|---|---|---|---|---|---|---|---|---|---|
| | $Score_{GT}$ | Upper RF | CI UL | $Score_{GT}$ | Upper RF | CI UL | $Score_{GT}$ | Upper RF | CI UL |
| **Extended Action Space** | | | | | | | | | |
| SP | 0.90 | 0.96 | 0.90 | 2.12 | 4.52 | 3.02 | 1.42 | 1.68 | 1.88 |
| SN | 5.82 | 6.40 | 10.48 | 2.68 | 3.98 | 4.54 | 9.44 | 11.10 | 11.55 |
| SPN | 0.90 | 0.96 | 0.90 | 1.25 | 2.06 | 2.86 | 1.23 | 1.47 | 1.31 |
| DN | 1.31 | 1.37 | 1.67 | 0.77 | 0.96 | 2.03 | 9.30 | 10.35 | 10.65 |
| CDN | 1.27 | 1.33 | 1.63 | 6.81 | 7.11 | 9.06 | 8.24 | 9.29 | 9.17 |

# E  EPISODIC REWARDS WITH CORRESPONDING SCORE$_{GT}$ FOR YAWNING TITAN EXPERIMENTS

Here we show the mean episodic rewards for each reward function, both action spaces, and all 3 agent orders. These results highlight the importance of our ground truth scoring method as it provides a common basis for comparing between agents trained using different reward functions. In addition, the poor correlation between mean episodic rewards and Score$_{GT}$ shows the need for better evaluation metrics in ACD.

Table 17: PPO Score$_{GT}$ and episodic mean rewards for all agent order combinations, averaged over all network sizes, for all 5 reward functions: sparse positive (SP), sparse negative (SN), sparse positive negative (SPN), dense negative (DN) and complex dense negative (CDN).

| Reward Function | Red then Blue | | Blue then Red | | Random | |
|---|---|---|---|---|---|---|
| | $Score_{GT}$ | Mean Episodic Reward | $Score_{GT}$ | Mean Episodic Reward | $Score_{GT}$ | Mean Episodic Reward |
| **Basic Action Space** | | | | | | |
| SP | 0.9 | 100.0 | 5.1 | 5.6 | 7.7 | 8.4 |
| SN | 8.0 | -10.8 | 9.6 | -3.0 | 12.2 | -0.1 |
| SPN | 0.9 | 75.0 | 1.1 | 7.5 | 3.9 | 9.5 |
| DN | 3.2 | -646.9 | 3.1 | -729.4 | 11.2 | -902.9 |
| CDN | 3.8 | -662.4 | 4.7 | -825.2 | 17.0 | -1001.6 |
| **Extended Action Space** | | | | | | |
| SP | 0.9 | 100.0 | 0.3 | 93.5 | 6.9 | 25.1 |
| SN | 9.3 | 0.0 | 9.0 | -2.6 | 12.5 | -0.5 |
| SPN | 0.9 | 80.0 | 0.6 | 69.0 | 4.5 | 20.6 |
| DN | 4.1 | -492.9 | 3.0 | -532.9 | 11.8 | -1029.5 |
| CDN | 5.7 | -657.8 | 4.0 | -664.4 | 17.6 | -1100.8 |

# F    POLICY ANALYSIS OF YT AGENTS

Table 18: The average blue action counts for each set of agents in an 100 step episode, averaged across network sizes, for the extended action space. These are the result of 1000 episodes of evaluation for each of the 25 agents in each set.

| Blue actions | SP | SN | SPN | DN | CDN |
|---|---|---|---|---|---|
| **Action Order: Red then Blue** | | | | | |
| Scan Network | 0 | 0.36 | 0 | 0 | 0 |
| Restore Node | 100 | 67.61 | 100 | 88.40 | 81.29 |
| Place Decoy | 0 | 32.03 | 0 | 11.60 | 18.71 |
| **Action Order: Blue then Red** | | | | | |
| Scan Network | 0 | 12.24 | 0 | 0.20 | 0.98 |
| Restore Node | 0.63 | 42.96 | 32.71 | 59.84 | 41.07 |
| Place Decoy | 99.37 | 38.07 | 67.29 | 39.96 | 57.96 |
| **Action Order: Random** | | | | | |
| Scan Network | 0.06 | 0.69 | 0.01 | 0.01 | 1.34 |
| Restore Node | 59.07 | 82.43 | 73.52 | 89.40 | 80.71 |
| Place Decoy | 40.87 | 16.88 | 26.47 | 10.59 | 17.94 |

# G POLICY ANALYSIS OF CAGE AGENTS

Here we show the detailed behaviour of agents trained in the MiniCAGE environment using sparse and dense rewards.

Table 19: In MiniCAGE, mean successful Impact action counts, and mean privilege red access counts, for each reward function. Evaluated over 1000, 100-step episodes.

| Reward Function | Impact Op Server Count | Op Server Privilege access Count | Enterprise host Privilege access Count | User host Privilege access Count |
|---|---|---|---|---|
| SP | 0.24 | **0.25** | **1.29** | **21.00** |
| SN | 4.53 | 10.18 | 2.28 | 1.06 |
| SPN | 2.25 | 3.57 | 2.76 | 22.21 |
| CDN (Default CAGE Rewards) | **0.02** | 7.89 | 22.22 | 4.50 |

Table 20: Mean blue action counts on each subnet and operational server for each reward function, evaluated over 1000 episodes (100 time steps). This table only includes the most relevant actions, with others like 'analyse' not included for conciseness.

| Action | SP | SN | SPN | DN (default CAGE rewards) |
|---|---|---|---|---|
| Decoy – User host | 2.15 | 5.26 | 1.64 | 1.40 |
| Decoy – Ent host | 4.94 | 4.91 | 5.43 | 1.81 |
| Decoy – Op server | 0.85 | 4.59 | 0.32 | 1.01 |
| **Decoy Total** | **7.94** | **14.76** | **7.39** | **4.22** |
| Remove – User host | 14.93 | 8.74 | 8.82 | 1.37 |
| Remove – Ent host | 1.47 | 3.59 | 1.87 | 2.21 |
| Remove – Op server | 0.09 | 1.14 | 0.18 | 0.38 |
| **Remove Total** | **16.49** | **13.47** | **10.87** | **3.96** |
| Restore – User host | 62.14 | 4.83 | 62.87 | 17.13 |
| Restore – Ent host | 3.22 | 4.91 | 5.34 | 56.10 |
| Restore – Op server | 0.17 | 14.46 | 6.45 | 12.32 |
| **Restore Total** | **65.53** | **24.19** | **74.66** | **85.55** |

# H CODEBASE

For the anonymised submission, a zipped folder containing the main codebase can be found in the submitted supplementary material.

# I   PPO Agent Training Curves in Yawning Titan

To accompany the detailed 50 node network data in Table 6, here we provide the training curves for each reward function and agent order in the extended action space.

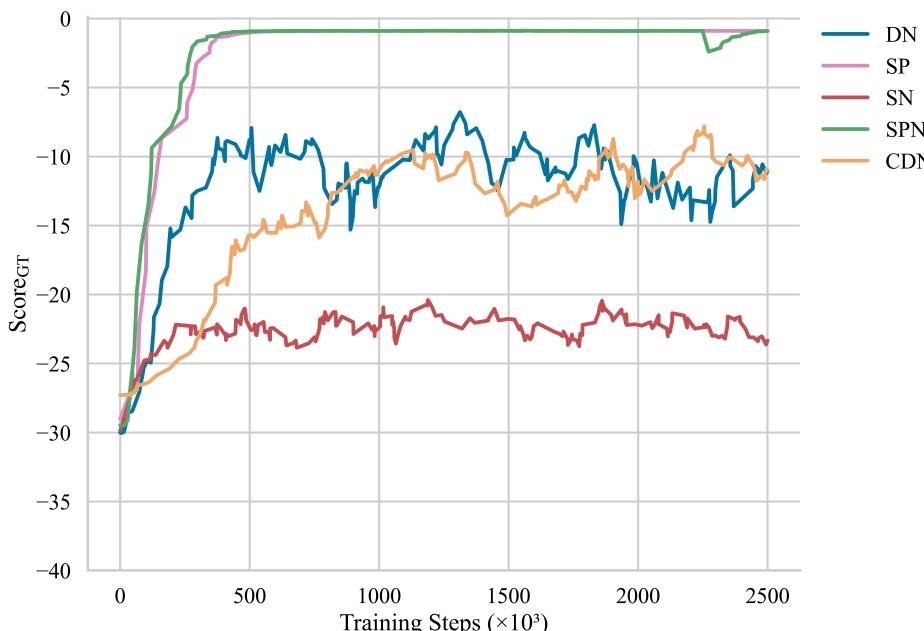

Figure 2: Training curves for the 50 node network size, red then blue agent order in the extended action space for reward functions: sparse positive (SP), sparse negative (SN), sparse positive negative (SPN), dense negative (DN) and complex dense negative (CDN).

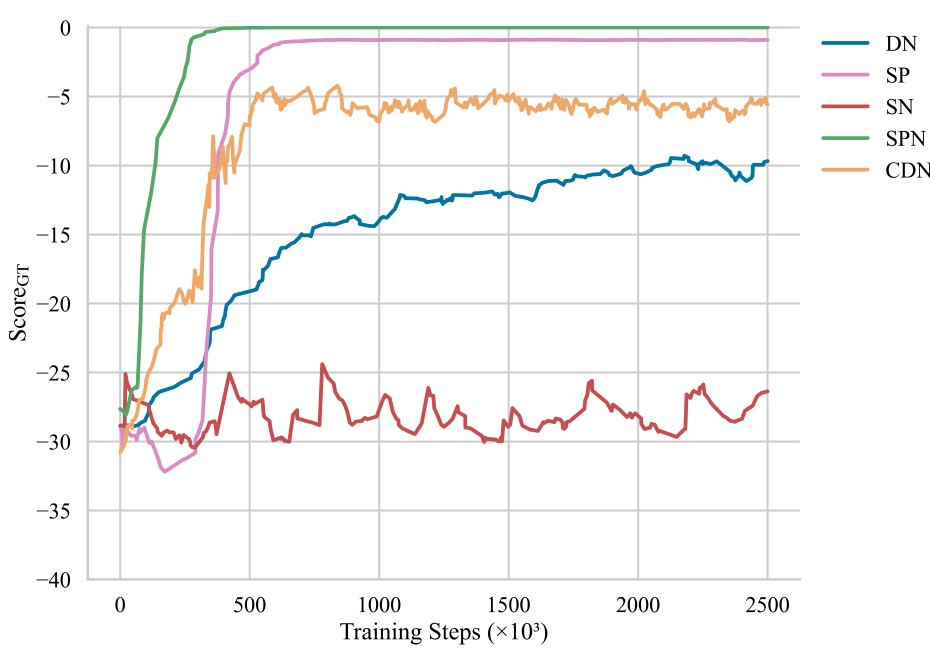

Figure 3: Training curves for the 50 node network size, blue then red agent order in the extended action space for reward functions: sparse positive (SP), sparse negative (SN), sparse positive negative (SPN), dense negative (DN) and complex dense negative (CDN).

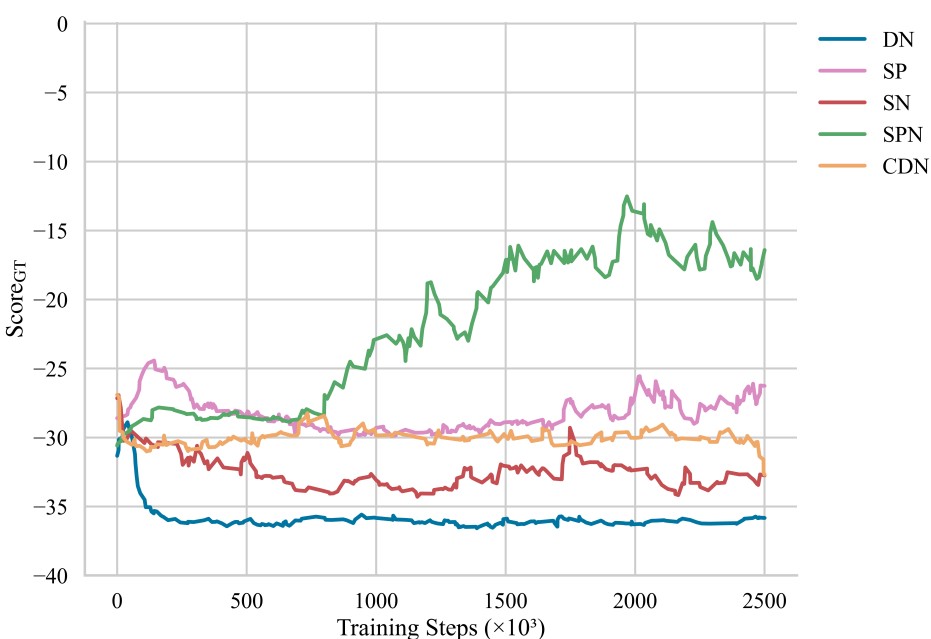

Figure 4: Training curves for the 50 node network size, random agent order in the extended action space for reward functions: sparse positive (SP), sparse negative (SN), sparse positive negative (SPN), dense negative (DN) and complex dense negative (CDN).

## J   CAGE AGENTS TRAINED USING DQN

Here, in Table 21, we detail our results from evaluating the sparse and dense reward functions in MiniCAGE using DQN.

Table 21: Results for DQN agents trained in MiniCAGE using the sparse positive (SP), sparse negative (SN), sparse positive negative (SPN) and default CAGE reward function (D).

| Reward Function | Score$_{GT}$ | Average Evaluation Reliability | | | | 95% CI | |
|---|---|---|---|---|---|---|---|
| | | Lower RF | Upper RF | $\bar{DT}$ (e-3) | DR$'$ | LL | UL |
| SP | 1.48 | 0.96 | 2.97 | 0.97 | 0.45 | 1.42 | 1.55 |
| SN | 2.75 | 1.78 | 3.65 | 0.01 | 0.28 | 2.67 | 2.84 |
| SPN | 1.51 | 0.97 | 2.85 | 0.97 | 0.40 | 1.45 | 1.58 |
| CDN (Default CAGE rewards) | 1.57 | 1.02 | 2.32 | 1.30 | 0.43 | 1.52 | 1.63 |

## K   POSITIVE REWARD ABLATION STUDY

Here we ablate the positive numerical sign from our SP reward to investigate the role of reward sign versus sparsity.

It is notable that rewards which perform poorly in our experiments (CDN, DN, SN) all feature negative penalties without any positive rewards. However, for an idealised RL algorithm the optimal policy is invariant when a constant is added to the reward function ( Sutton & Barto (2018)). To empirically determine the role that numerically positive rewards play in the improved performance of SP and SPN agents (i.e., versus sparsity) we create the Ablated-SP reward which simply adds a constant reward (-1) to the SP reward. In other words, the blue agent receives a reward of 0 when the network has zero compromised nodes and -1 otherwise.

Using the in the YT environment, a network size of 10 nodes and all three agent orders, we train each agent for 1.5 million time steps and evaluate the Score$_{GT}$, best Score$_{GT}$ and number of optimal runs.

Table 22: YT PPO agent results for each agent action order combination in the basic and extended action spaces for network of size 10, comparing agents trained using the Ablated-SP reward function with the alternatives.

| | 10 Node Network Evaluation | | | | | | | | |
|---|---|---|---|---|---|---|---|---|---|
| | Red then Blue | | | Blue then Red | | | Random | | |
| Reward Function | Score$_{GT}$ | Best Score$_{GT}$ | No. of Optimal Runs (/25) | Score$_{GT}$ | Best Score$_{GT}$ | No. of Optimal Runs (/25) | Score$_{GT}$ | Best Score$_{GT}$ | No. of Optimal Runs (/25) |
| **Basic Action Space** | | | | | | | | | |
| SP | 0.90 | 0.90 | 25 | 3.91 | 0.90 | 10 | 1.35 | 1.34 | * |
| Ablated-SP | 0.90 | 0.90 | 25 | 2.61 | 0.90 | 20 | 1.35 | 1.34 | * |
| SN | 6.15 | 0.90 | 3 | 6.10 | 0.9 | 2 | 8.35 | 7.62 | * |
| SPN | 0.9 | 0.90 | 25 | 0.9 | 0.9 | 25 | 1.35 | 1.34 | * |
| DN | 1.57 | 0.90 | 12 | 1.76 | 0.9 | 15 | 6.85 | 1.34 | * |
| CDN | 2.38 | 0.90 | 5 | 1.53 | 0.9 | 12 | 8.79 | 8.54 | * |
| **Extended Action Space** | | | | | | | | | |
| SP | 0.90 | 0.90 | 25 | 0 | 0 | 25 | 0.99 | 0.88 | * |
| Ablated-SP | 0.90 | 0.90 | 25 | 0 | 0 | 25 | 1.22 | 1.18 | * |
| SN | 7.39 | 2.87 | 0 | 6.84 | 0 | 1 | 8.18 | 6.73 | * |
| SPN | 0.9 | 0.9 | 25 | 0.6 | 0 | 18 | 1.04 | 0.88 | * |
| DN | 1.82 | 0.9 | 7 | 1.48 | 0 | 4 | 6.33 | 1.34 | * |
| CDN | 2.71 | 0.9 | 5 | 2.39 | 0 | 3 | 6.98 | 1.35 | * |

* The optimal policy score is non-trivial so we do not count the number of optimal runs

The results in Table 22 show the Ablated-SP reward function (which does not include any positive rewards) achieves the same (or better) average $Score_{GT}$ as the SP reward in the two fixed agent orders. When the agent order is random, Ablated-SP outperforms the CDN, DN and SPN rewards, and closely approaches the performance of SP and SPN – likely due to hyperparameter sensitivity during training. This shows that including numerically positive rewards is not the main reason that SP and SPN outperform SN, DN and CDN.

## L    DEFAULT CAGE-2 REWARD SOURCES OF BIAS

The CAGE 2 reward function is dense, highly-engineered and contains potential sources of bias that may lead to misaligned or sub-optimal (e.g., because of a noisy or contradictory reward signals) policies. In contrast to sparse rewards, it is also highly tailored to the specific CAGE-2 challenge scenario and therefore unsuited to modified network configurations without additional work. The specific sources of bias in the CAGE 2 reward function are as follows:

1. All compromised user hosts provide the same penalty despite having different vulnerability profiles (and thus different long-term state values). The same is true of hosts in the enterprise subnet.

2. The penalty for enterprise hosts and operational server compromise is the same (-1) despite a compromised operational server being much closer to an impacted operational server (-10) from a lateral movement (causal distance from attack objective) perspective.

3. The penalty for a compromised operational host is -0.1 per time step, the same as a user host, despite requiring fewer steps to reach and impact the operational server. This is an example where the default reward would not generalise well to an adversary that made use of this route, yet a sparse reward would not require modification or domain expertise.

4. The cost of performing the restore action is -1, drawing numerical equivalence with the compromise of an enterprise or operational host. It is also equivalent to the compromise of a user host for 10 steps. This means that the resulting policy is biased towards restoring only enterprise or operational hosts (as seen in Table 20). This may yield conflicting signals with the fact that compromising user hosts is causally necessary for impacting the operational sever, thus failing to restore them leaves the adversary closer to operational impact. Supporting this hypothesis, the SP and SPN rewards use the restore action more sparingly overall and use it mainly on user hosts.

## M    MiniCAGE Evaluation using the Default CAGE-2 Reward

In Table 23 we show the mean and median scores per timestep using the default CAGE 2 reward across the sets of policies trained using sparse reward functions. The results show that SPN performs similarly to CDN in terms of the mean score (-1.01 vs -0.99), and that SP and SPN perform better in terms of the median (-0.96/7 vs -1.09). We include GT score for comparison and the results are also supported by the agent policy analysis in Appendix G.

These results can be understood further by considering Table 3 which shows the upper RF of SP and SPN rewards is higher than CDN i.e., the worst 5% of policies have lower scores. This is because there is a higher probability that the operational server is impacted and incurs a large negative penalty. Since operational server impact is causally dependent on user and then enterprise host compromise, and our sparse policies do a much better job of confining adversary impact to the user hosts, we think this may be an exploration issue that could be solved with further hyperparameter tuning. Alternatively, it may be that the optimal way to defend the op server at all costs is by sacrificing enterprise hosts - keeping the adversary 'stuck' near the target rather than minimising overall network compromise. This seems untenable for real-world cyber defence.

Table 23: The MiniCAGE agents evaluated over 1000 episodes (one episode is 100 steps) using the $Score_{GT}$ and the original CAGE reward function averaged (Mean and Median) over each timestep.

| Reward Function | $Score_{GT}$ | Mean score per timestep using CAGE 2 default reward | Median score per timestep using CAGE 2 default reward |
|---|---|---|---|
| SP | 1.29 | -1.37 | -0.97 |
| SN | 2.77 | -2.25 | -2.04 |
| SPN | 1.35 | -1.01 | -0.96 |
| CDN (CAGE 2 Default) | 1.41 | -0.99 | -1.09 |

