# OpenReview forum: "Rewards Simplified: Reducing Risk in RL for Cyber Defence"
_ICLR.cc/2026/Conference — Submitted to ICLR 2026_

### Official Review · Reviewer_dku5 · 2025-10-17

**Soundness:** 2
**Presentation:** 3
**Contribution:** 2
**Rating:** 2
**Confidence:** 3

**Summary:**

In the area of autonomous cyber defence, this paper provides an evaluation of various choice of reward functions, arguing that sparse rewards provide for improved performance and reliability. The paper conducts experiments in two established cyber-gym environments, and considers the effects of network size, etc. to the evaluation. As part of their approach, a scoring method is used that it is claimed to address some of the shortcomings of extant approaches.

**Strengths:**

The paper is well written in general. The experiments conducted are reasonably comprehensive in their motivated scope (though with shortcomings noted further down).

The arguments for the change to the ground truth scoring used and the reliability evaluation chosen are clear.

**Weaknesses:**

One of my main concerns re this paper may stem from a misunderstanding. As such, I am happy to increase my score if it is established as such. But the results seem to suggest that consistently SP and SPN perform better than SN, DN and CDN. Naively to me, this indicates that postive-focused rewards functions could actually be the distinguishing beneficial feature rather than sparse rewards necessarily. Further, it seems to be that SN is worse than DN, which is not supporting of that fact that sparseness is the winning factor in the rewards formulation.

It would have been good to see results also for older metrics which the authors have built upon to come up with the metrics they used, i.e. in order to see the benefit of the improved metrics.

Overall, the proposed metrics and rewards are relatively simple. The main contribution of the paper is incremental, and the evaluation of these incremental changes is potentially not comprehensive enough or conclusive.

**Questions:**

I would appreciate the authors' comments on the question above as to why the results show that e.g. sparseness rather than, say, positivity is the key element of an improved rewards function. And, of course, any other responses to the potential weaknesses noted would be appreciated.

---

> ### Author Response · Authors · 2025-11-28
>
> We thank the reviewer for their questions and their acknowledgement of the comprehensive experimental work contained in the paper. We respond to your comments and questions below:
>
> - **Q1. Why do SP and SPN perform well while SN performs poorly? Is positive sign rather than sparsity the determining factor?**
>
> Thank you. We think this is similar to Q1.2 from Reviewer 1 and now clarify this further in the paper. We do not believe that sign determines the poor performance of SN, DN and CDN. We now evaluate this using an ablation of the SP reward in Appendix K. It shows that adding the constant -1 to the SP reward (thus 0 for no-compromises and -1 otherwise)  still outperforms dense rewards for the 10 node network. This is supported by RL theory e.g., adding a constant to all states should not change the value function in an idealised RL algorithm (given as an exercise in Sutton and Barto, 2008). We will add to these ablations for the final version.
>
> We do also note that, although adding a constant does not change the optimal policy wr.t idealised RL, changing the sign does impact DRL algorithms including PPO and DQN in practice. e.g., PPO’s GAE estimation interacts poorly with the larger episodic negative rewards (e.g., ~-100 vs 1), causing inflated policy gradient losses and a larger proportion of clipped updates during the early training. We compensate for this with hyperparameter tuning but recognise this may warrant further study in future.
>
> - **Q2. SN being worse than DN seems to contradict the claim that sparsity is beneficial.**
>
> We agree, however our claim (which we have now clarified) is that sparse rewards are beneficial only when they are (goal-aligned and) encountered with sufficient frequency (i.e., >0 until policy convergence).  The poor performance of SN stems from extreme sparsity. As the network size increases, and the policy becomes better at defending the network, the agent receives little-to-no reward signal. SP and SPN work because they are unbiased, goal aligned (node compromise is causally related to op server impact) and can be encountered frequently during training. We think that under different environment dynamics, e.g., a more powerful or dynamic adversary, and with additional hyperparameter tuning, SN rewards can still be useful as it does not suffer bias and is goal aligned (albeit less generically as the goal is preventing op server impact rather than the causal factors which generalise to more than the goal of preventing op server impact).
>
> - **Q3. Why not report results for older metrics the paper builds upon?**
>
> Thank you, we agree this is useful context. We have now added results comparing our sparse rewards (SP, SPN and SN) to the original reward for CAGE in Table 23, Appendix M. It shows SP and SPN rewards do indeed perform similarly to CDN even when evaluated using CDN. This is also supported by the agent policy analysis in Appendix G.
>
> - **Q4. Are the metrics and reward functions too simple or incremental?**
>
> To the best of our knowledge, this paper is the first systematic study to show that reward design can introduce significant and previously unrecognised risks in RL-based cyber defence. The rewards we propose are simple, but the key contribution of our paper is to show that there are quantifiable risks that emerge from the highly-engineered reward structures used in leading cyber gyms (CAGE and YT). This is only possible because we introduce Score$_{GT}$, identify the intra-step compromises missing from the abstractions in these environments, and then thoroughly evaluate using risk and reliability metrics that have not been applied in cyber gym evaluations before.
>
> To summarise the breadth of our experimental coverage includes (by far SoTA in the domain of RL for cyber defence): two environments, multiple network sizes, PPO and DQN, multiple attacker behaviours, three agent orderings, detailed analysis of the action distributions arising from different rewards, ablations for the reward sign and identifying biased components of the standard CAGE2 reward.
>
> These results challenge core assumptions in current cyber‑gym design and provide new metrics essential for safe and reliable RL-based cyber defence. We believe our results are significant and in a domain that is of critical importance.

---

### Official Review · Reviewer_uXAQ · 2025-10-24

**Soundness:** 2
**Presentation:** 4
**Contribution:** 3
**Rating:** 4
**Confidence:** 4

**Summary:**

The authors present a study on the importance of specific considerations for the reward function in the Yawning Titan and MiniCAGE cyber gyms. They propose a ground truth score to better capture intra-step compromise during episodes and use this to measure the effectiveness of sparse and dense reward functions. In addition to ground truth score, the authors also measure how varied reward functions affect reliability during training, and risk after training. Their experiments showed that sparse reward functions achieved better scores, better reliability, and lower risk across different environment sizes, action-spaces, and agent orderings when compared to dense reward functions.

**Strengths:**

* The use of the MiniCAGE environment strengthens the argument for this method’s utility across varied environments of growing complexity.
* Varied step order is another interesting area that the authors explore. Allowing the red agent to sometimes take 2 steps in a row is comparable to uncertain timing if agents were brought from the discrete world of simulation to a real-world system.
* The structure of the paper and the writing is strong. All experiments are repeatable from the body of the text.

**Weaknesses:**

* Using a line graph for Yawning Titan is overly simplistic. Making the line longer introduces very little additional complexity as the attack path is still the same, just longer.
* Though the CDN function that comes with the environment may be suboptimal for agent training, it is also the metric by which we score models. In the CAGE environment, certain hosts like the OpServer are more important than others, certain actions have associated costs. And while the inclusion of Tables 18 and 19 in Appendix F help to show that SP and SPN do indeed have lower OpServer impacts, and use fewer Restore actions, we already have a “Ground truth” reward function for CAGE: the environment’s default reward function. The ground truth function should reflect the goals of the environment designers; if all hosts were weighted the same, that would be reflected in the reward function. The authors could make a much stronger argument for their method by showing that using SP or SPN during training results in better scores from the environment’s reward function directly (which appears to be the case). If this is not the case, then they are optimizing their agents for an entirely different goal; it is unclear why one should expect an agent trained to maximize “goal 1” would also maximize “goal 2” if the two goals are not strongly correlated.
* There should be an additional policy analysis of the YT agents. It is unclear if the CDN agent is more “reluctant” to use the restore and decoy actions because they have associated costs, leading to lower ground truth scores.


Minor issues/nitpicks:
* I disagree with the authors’ characterization of the B-Line agent as the harder agent in Section 3.2. The leaderboard for the original CAGE-2 contest shows Meander was the most difficult agent for submissions.
* Please bold the best values in Tables 2-6
* The authors state on numerous occasions that “sparsely rewarded polices perform better”, but this is only the case for SP and SPN. SN is consistantly the worst policy evaluated.
* Sec 3.2: The “A” in CAGE stands for “autonomous” not “autonomy”.

**Questions:**

* How does each reward function compare wrt the environments’ original reward functions? If they perform worse according to the default reward function, why should we say that the global reward is a better metric? If they perform better, why?
* Are agents trained using only red-blue action ordering before being evaluated in the different environments? Would varying move order during training have a stronger effect on agent policy than the choice of reward function?
* Is it possible to evaluate agent action order in the MiniCAGE environment as well?

---

> ### Author Response · Authors · 2025-11-28
>
> Thank you for noting the strength of our methodology and writing structure. We address your questions below:
>
> - Q1.1. How does each reward function compare wrt the original reward functions?
>
> We now evaluate this for the set of reward functions (and default CDN) in CAGE 2, as per Table 23 added to Appendix M. It shows SPN performs similarly to CDN in terms of mean score (-1.01 vs -0.99), and that SP and SPN perform better in terms of the median (-0.96/7 vs -1.09).
>
> These results can be understood further considering Table 3 showing the upper RF of SP/SPN is higher than CDN. This is because of a higher probability that the op server is impacted which incurs a large negative penalty. Since op server impact is causally dependent on user and then enterprise host compromise, and our sparse policies do a better job of confining adversary impact to user hosts, we think this could be solved with further hyperparameter tuning (I.e. exploration). Alternatively, it may be the optimal way to defend the op server is by sacrificing enterprise hosts - keeping the adversary ‘stuck’ near the target rather than minimising overall network compromise. This seems untenable for real-world cyber defence.
>
> - Q1.2. If they perform worse according to the default reward, why should we say that the global reward is a better metric? If they perform better, why?
>
> We believe SP/SPN perform well because they avoid action and state penalties that may conflict in a way which antagonises the overarching goal of preventing compromised nodes. CDN should perform best against the CAGE 2 default reward function because the optimisation target and evaluation score are the same.
>
> SPN optimises the ground truth score and performs nearly as well (-1.01 vs -0.99) as CDN agents when evaluated using CDN. Therefore Score$_{GT}$ does not contradict the designer’s intent, and neither is the point of our paper to design the best possible reward for CAGE 2, instead it is an important evaluation tool that reveals previously hidden risks and allows evaluating a range of rewards for a given goal.
>
> - Q2.1. Are agents trained using only red-blue action ordering before being evaluated in the different environments?
>
> No, The agents are trained and evaluated using each agent order (e.g., trained in blue-red and evaluated in the blue-red) - this is now clarified in Section 3.5.
>
> - Q2.2. Would varying move order during training have a stronger effect on agent policy than the choice of reward function?
>
> Shown in Table 5, agent order has a large impact on environment dynamics. A key finding is that the sparse rewards perform well despite agent order. This is because they are not excessively tailored to the specific scenario and, for a given MDP, focus on host compromise which is causally related to the operational objective. We would not say the effect of agent order is stronger, but something that cannot be fully anticipated and gives reason to avoid highly engineered rewards.
>
> - Q3. Is it possible to evaluate agent action order in the MiniCAGE environment as well?
>
> Thank you, this is planned future work but we have not yet produced these results. CAGE 2 (thus MiniCAGE) used a fixed agent order and we focussed on generalising to the larger state-action space rather than adding this functionality.
>
> **Additional points r.e.,Minor issues:**
>
> - I disagree with the authors’ characterisation of the B-Line agent
>
> This is a good point and we have adjusted our language in the paper. At the time of submission only B-Line was included in MiniCAGE. B-Line is in principle the harder adversary since it has prior knowledge about the network and moves more efficiently (taking the “beeline”) to the op server. In practice agents do better against B-Line because, we suspect, partial observations are overwhelmed by the greater variety of agent behaviour in meander. We have since added Meander to MiniCAGE and will aim to include those results in the final paper.
>
> - Bold the best values in Tables 2-6
>
> Done.
>
> - The authors state that “sparsely rewarded polices perform better”, but this is only the case for SP and SPN. SN is consistently the worst policy evaluated
>
> Thank you. We since clarify that only sparse rewards that are both goal-aligned and encountered with sufficient frequency perform better in terms of Score$_{GT}$. The main issue with SN is that it is not encountered frequently, and as training progresses the issue is exacerbated. A policy that avoids complete compromise but falls short of compromise-free ceases to receive any reward signal. We think that under different environment dynamics, e.g., a more powerful or dynamic adversary, and with additional parameter tuning, SN rewards would still be useful as they do not suffer bias and are goal aligned.
>
> - Sec 3.2: The “A” in CAGE stands for “autonomous” not “autonomy”
>
> The original CAGE paper states “Cyber Autonomy Gym for Experimentation (CAGE)”, see arXiv 2309.07388. Let us know if this has been superseded we are happy to fix it.

---

### Official Review · Reviewer_xvQ3 · 2025-10-29

**Soundness:** 2
**Presentation:** 3
**Contribution:** 3
**Rating:** 4
**Confidence:** 4

**Summary:**

This paper evaluates the use of sparse and dense reward functions in two cyberattack-defense gym environments, YT and CAGE 2. The motivation is to understand whether dense rewards may introduce bias during training and whether sparse rewards can improve effectiveness and reliability while reducing risk. The authors evaluate multiple sparse and dense reward functions, including the standard dense reward functions used over multiple network sizes and different agent orders. The results show that sparse rewards lead to better performance and reliability.

**Strengths:**

The paper makes an interesting observation that the dense, highly engineered rewards in cyber gym environments used for training autonomous cyber defence (ACD) agents may be misleading and limit the agents' performance. Although similar observations have been made in other RL settings, such as training board game agents, a systematic study of sparse rewards in the ACD settings was lacking. The evaluation using the YT and CAGE environments, including their standard dense rewards, is convincing. These results reveal flaws in the design of reward functions in current ACD gym environments, making a valuable contribution.

**Weaknesses:**

1. While the observations are interesting, a deep analysis of the results is lacking. In particular, it would be interesting to identify which components in the dense reward functions lead to low scores and instability in YT and CAGE. For example, for YT, it seems that all dense rewards only provide negative signals, but no positive signals. I wonder if that's why the dense rewards perform worse than SPN and SP. This is also consistent with the fact that SN is often the worst. From that perspective, dense rewards actually perform better than SN by providing more detailed feedback. Hence, the conclusion that sparse rewards are always better is questionable even for the relatively simple scenarios considered in the paper. Their generalization to other settings is even less unclear.
2. The results were obtained assuming simple attack strategies. I wonder if similar observations hold for more sophisticated attacks, e.g., when the attacker also uses RL.
3. The paper identifies a problem with current cyber gym environments, where agents cannot "reliably distinguish between states in which nodes have been compromised and those in which no compromise occurred." I wonder if that affects the simulation results. That is, can one modify the simulators to provide the correct reward signals to the agents?

**Questions:**

Please see the discussions on weaknesses above.

---

> ### Author Response · Authors · 2025-11-28
>
> We would like to thank the reviewer for their helpful comments and recognition that our evaluation using YT and CAGE environments, including standard dense rewards, is convincing and makes for a valuable contribution. We address your questions below:
>
> **Rewards (1)**
>
> - *Q1.1. Which components of dense reward functions cause low scores and instability in YT and CAGE?*
>
> Good question. We designed our methodology to cover rewards ranging from extremely sparse to highly-engineered (i.e., CAGE 2). One way in which bias arises in the CAGE-2 reward (thus lower Score$_{GT}$ and instability) is the penalty for enterprise hosts and op server compromise is the same (-1), despite a compromised op server being much closer to an impacted op server (-10) from a lateral movement (~distance from attack objective) perspective.
>
> The overall point we strive to make is that hand-engineering complex reward structures with domain knowledge is likely to cause policy bias in non-trivial ways. We now discuss this in Appendix L.
>
> - *Q1.2. For YT dense rewards, are the signals only negatives? Does this explain poor performance?*
>
> Thank you for raising this. It is true that the YT dense reward contains only penalties however we do not believe this predominantly determines the poor performance. The optimal policy is invariant under adding a constant to all states (given as an exercise in Sutton and Barto 2008), so adding -1 to the SP reward i.e., providing 0 for zero-compromise and -1 otherwise should yield equivalent policies.
>
> We now investigate this in Appendix K by ablating the numerical positive in SP and show the Ablated-SP reward still outperforms dense rewards for the 10 node network. We can add a more complete set if the reviewer believes it would benefit the final version. e.g., we would like to show that the equivalent “dense positive” reward is also biased.
>
> We note that, although adding a constant does not change the optimal policy w.r.t idealised RL, changing the sign does impact DRL algorithms including PPO and DQN in practice. e.g., PPO’s GAE estimation interacts poorly with the larger episodic negative rewards (e.g., ~-100 vs 1), causing inflated policy gradient losses and a larger proportion of clipped updates during the early training. We compensate for this with hyperparameter tuning but recognise this may warrant further study in future.
>
> - *Q1.3. Does SN being worst mean dense rewards are better than SN and thus the conclusion that sparse is better is questionable?*
>
> We believe the poor performance of the SN reward is mainly attributable to the infrequency with which it is encountered - a problem that is exacerbated as network size grows as shown in Table 4.
>
> By only penalising the complete compromise of the network, policies that prevent complete compromise but fall short of zero compromise do not receive any reward signal.  During the initial stage of policy learning, the reward signal drops to nearly zero and they stop improving. We think that additional hyperparameter tuning may alleviate this somewhat, as might increasing the size of the penalty to strengthen the advantage signal, but the issue of encountering the reward too infrequently makes it much less efficient than rewards conditioned on zero-compromise.
>
> - *Q1.4. Is the conclusion “sparse rewards are always better” overstated?*
>
> We agree such a universal claim would be overstated, and we have adjusted the text to clarify where our contributions lie. i.e., our results show that sparse goal-aligned rewards that are encountered frequently are better than dense rewards in YT and CAGE across the settings we evaluate. Our findings concern the structure and frequency of rewards relative to cyber-defense goals and sign is not the primary variable of interest. Nevertheless we now investigate this via ablation in Appendix K and show that the sign can be changed whilst maintaining better performance than the dense rewards.
>
> **Scope (2)**
>
> - Q2. Would these results hold under more sophisticated attackers (e.g., RL-based)?
>
> Thank you. This is beyond the scope of our paper which focusses on real-world cyber gyms, and these don’t include this functionality. We note that by varying the agent order we do partially explore the impact of varying attacker dynamics (e.g.., SP/SPN remain superior even when the attacker takes two consecutive turns in the randomised order).
>
> **Environment Design (3)**
>
> - Q3.1. Does the inability to observe intra-step compromise affect simulation results?
>
> Absolutely. We identify that these existing environments do not observe the intra-step compromise and thus miss important attacker behaviours (especially as their rewards - and thus evaluations - are not sensitive to intra-step). Our Score$_{GT}$  mechanism addresses this and also shows that sparse rewards do better despite lacking this context.
>
> Q3.2. Can simulators be modified to provide correct reward signals to agents?
>
> Yes - we will release this with our source code.

---

### Meta-Review · Area_Chair_Uutd · 2025-12-27

**Summary:**

The present paper is focusing on reinforcement learning based approaches for training cyber defence agents. It evaluates different algorithmic design choices in this setting including reward structure, network sizes, as well as policy and value-based RL algorithms focusing on the YT and CAGE 2 gym environments. In these environments, the RL agent acts as a high-level actor overseeing a network under attack and can take actions to defend against the attack (such as analyzing and reseting individual macines in the network).

For their evaluation, the authors propose a novel ground truth score to avoid what they call the "intra-step compromise". This is an issue with the reward computation in such environments, where no cost is incurred for, as I understand it, having fixed a node in the same timestep as it was attacked. The main finding of the work is that sparse rewards yield better policies. They are better aligned with the defenders goals and use fewer costly defensive actions.

**Reviewer Concerns:**

The reviewers raised several concerns on simplicity, including attack strategies (xvQ3), metrics and rewards (dku5). Some other concerns were more technical in nature and requested further analysis (uXAQ). Probably the most important concern has been the question about the negativity of the reward values for dense rewards and its impact on the policy (xvQ3,dku5).

**Reviewer Scores:**

The authors addressed most of the concerns likely to the reviewers' satisfaction. The main exception to this is the question about the negativity of the reward values. In their response, the authors rightly point out that adding a constant to the rewards should, in theory, not impact the resulting policy. They even show this experimentally, showcasingthe relative order of agent performance to remain the same. At the same time, the authors admit that "changing the sign does impact DRL algorithms, including PPO and DQN, in practice," and rightly say that this warrants further study. Given that this phenomenon lies at the core of what this paper seeks to investigate, I believe that the reviewers may find the authors' response not convincing enough to raise their scores towards acceptance.

---

### Decision · Program_Chairs · 2026-01-26

Reject